# One-Shot Object Detection
# with Co-Attention and Co-Excitation

**Ting-I Hsieh**[1], **Yi-Chen Lo**[1], **Hwann-Tzong Chen**[1,3], **Tyng-Luh Liu**[2,4]
[1]National Tsing Hua University, [2]Academia Sinica, Taiwan, [3]Aeolus Robotics, [4]Taiwan AI Labs
{tingihsieh.tw,howardyclo}@gmail.com, htchen@cs.nthu.edu.tw, liutyng@ailabs.tw

## Abstract

This paper aims to tackle the challenging problem of one-shot object detection. Given a query image patch whose class label is not included in the training data, the goal of the task is to detect all instances of the same class in a target image. To this end, we develop a novel *co-attention and co-excitation* (CoAE) framework that makes contributions in three key technical aspects. First, we propose to use the non-local operation to explore the co-attention embodied in each query-target pair and yield region proposals accounting for the one-shot situation. Second, we formulate a squeeze-and-co-excitation scheme that can adaptively emphasize correlated feature channels to help uncover relevant proposals and eventually the target objects. Third, we design a margin-based ranking loss for implicitly learning a metric to predict the similarity of a region proposal to the underlying query, no matter its class label is seen or unseen in training. The resulting model is therefore a two-stage detector that yields a strong baseline on both VOC and MS-COCO under one-shot setting of detecting objects from both seen and never-seen classes. Codes are available at https://github.com/timy90022/One-Shot-Object-Detection.

## 1 Introduction

The ability of humans to learn new concepts under limited guidance is remarkable. Take, for example, the task of learning to identify and localize a never-before-seen object in an image based on a given query template. Even without prior knowledge about the object's category, the human visual system has evolved to be able to handle such a task by performing different functionalities that include grouping the pixels of objects as a whole, extracting distinctive cues for comparison, and exhibiting attention or fixation for localization. All these can be done under a wide range of variations in object appearances, viewing angles, lighting conditions, and so on.

The goal of this work is to address the problem of one-shot object detection by taking account of achieving the aforementioned capability and flexibility of the human visual system when a similar one-shot task of perceptual categorization and localization is performed. We assume that a query image of an object will be provided as an exemplar or a prototype of some unseen class, and the task is to localize the most likely occurrences of the query object in a new target image. Further, we require that the query object must not belong to any seen class at any level in the categorical hierarchies during training. It is also worth noting that the definition of object category may vary over different context [1]. The contextual information for our one-shot scenario of object detection comes in two forms. First, the target image provides the spatial context, implying the likelihood of observing an object at a specific location with respect to the background and other foreground objects. Second, the query image and the target image jointly provide the categorical context. The exact level in the categorical hierarchies that both the query and the target objects belong to is determined by how they share significant numbers of attributes (such as color, texture, and shape) in common.

Metric learning is often employed as a key component to solve one-shot classification problems. However, it is not straightforward to apply a learned metric to one-shot object detection. The detector still needs to know which candidate regions in the target image are more likely to contain the objects to be compared with the query object using the learned metric. We propose to extract region proposals from non-local feature maps that incorporate co-attention visual cues of both the query and target images. On the other hand, object tracking can be considered as a special case of one-shot object detection with the temporal consistency assumption. The initial bounding box specified in the first frame can be viewed as the query. The subsequent frames are target images. A key difference between object tracking and our formulation of one-shot detection is that we do not assume that the target image must contain the same instance as the query image. It is allowed to have significant appearance variations between the objects, as long as there exist some common attributes for characterizing them as the same category. We present a new mechanism called *squeeze and co-excitation* to simultaneously emphasize the features of the query and target images for detecting objects of novel classes. The experiments show that our co-attention and co-excitation (CoAE) framework can better explore the spatial and categorical context information that is jointly embedded in the query and target images, and as a result, yields a strong baseline on one-shot object detection.

## 2    Related work

**Object detection**    State-of-the-art object detectors unanimously adopt variants of deep convolutional neural networks as their backbones and have been improving the performance on large-scale benchmarks. Two types of pipeline designs are often taken into consideration by recent object detectors: one-stage (proposal-free) [2–7] and two-stage (proposal-based) [8–13]. Two-stage detectors generate a set of region proposals at the first stage, and then classify the proposals as well as refine their locations at the second stage. The two-stage pipeline is first demonstrated by R-CNN [11] and further improved by Faster R-CNN [13], which replaces the grouping-based proposal method with a region proposal network (RPN), making the whole pipeline end-to-end trainable. Subsequently, state-of-the-art two-stage object detectors [8, 10] mainly follow Faster R-CNN in the design of architecture. In contrast, one-stage detectors like [2–7] trade localization performance for fast inference speed by skipping the region-proposal step and directly predicting the bounding boxes and the corresponding class labels with respect to a fixed set of anchors.

**Few-shot classification via metric learning**    The aim of metric-learning based few-shot classification is to derive a similarity metric that can be directly applied to the inference of unseen classes supported by a set of labeled examples (*i.e.*, *support set*). The setting of $N$-*way* $K$-*shot* classification is considered to have a support set containing $K$ labeled examples for each of $N$ classes, where $K = 0$, $K = 1$, and $K > 1$ mean zero-shot, one-shot, and few-shot, respectively. Koch [14] presents the first principled approach that employs Siamese networks for one-shot image classification. The Siamese networks learn a general similarity metric from pairs of input images to decide whether the two images belong to the same class. Then, during inference, the Siamese networks can be used to match unlabeled images of either seen or unseen classes with the one-shot support set. The prediction is done by assigning the test image with the class label of the most similar example in the support set. Vinyals et al. [15] propose the matching networks and an episodic training strategy tailored to the few-shot criterion. The matching networks learn a more powerful similarity metric using an attention mechanism [16, 17] over the examples in the support set. Instead of associating unlabeled samples with their nearest support examples, the prototypical networks proposed by Snell et al. [18] map the unlabeled samples to the nearest 'class prototype' of the support set. Snell et al. also show that the prototypical networks can be applied to zero-shot setting where the class prototype becomes the semantic vector of the class. Sung et al. [19] present the relation network, which is similar to [14, 15, 18] but learns the similarity metric fully based on a relational convolutional block instead of the Euclidean or cosine distance.

**Few-shot object detection**    Similar to few-shot classification, the problem of object detection can also be addressed under a few-shot setting. This problem is relatively new and less explored, and only a few preliminary results are reported from the perspectives of transfer learning [20], meta learning [21], or metric learning [22–24]. For transfer learning, Chen et al. [20] present the regularization techniques to address overfitting when directly training on a handful of labeled images of unseen classes. For meta learning, Kang et al. [21] propose a meta-model that is trained to reweight the

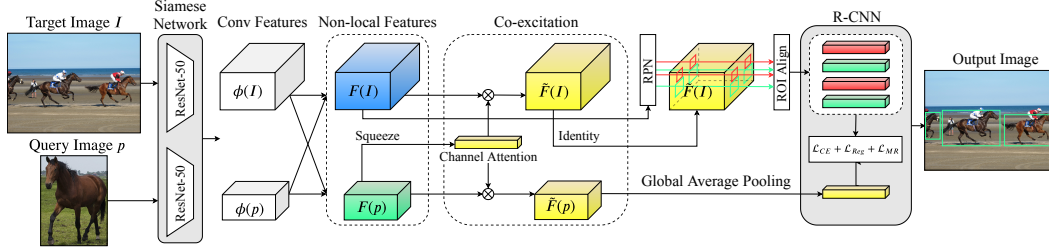

Figure 1: The overall neural network architecture of the propose method for one-shot object detection.

features of an input image extracted from a base detection model. The reweighted features can be adapted to the detection of novel objects from a few examples. For metric learning, [22–24] share a similar framework that replaces the conventional classifier in the object detector with a metric-based classifier akin to [14, 15, 18, 19].

The problem formulation of our work is more closely related to [14, 15, 18, 19, 22, 24] than [20, 21, 23]. Our formulation is class-agnostic and training-free for unseen novel classes. Once the training process is done, our model can be used to detect objects of unseen classes without either knowing the classes beforehand or the need of fine-tuning.

## 3   Our method

Consider the task of object detection over a set of class labels, denoted as $C$. As our method is designed to deal with the one-shot scenario, we further divide the label set by $C = C_0 \cup C_1$, where the former includes those are available during training and the latter comprises the remaining for the inference of one-shot object detection. We choose to tackle the one-shot detection task in two stages, and develop the proposed network architecture based on Faster R-CNN [13]. In our implementation we have experimented with using ResNet-50 as the CNN backbone and carried out extensive comparisons with other relevant techniques.

We formulate the one-shot object detection as follows. Given a query image patch $p$, depicting an instance of a particular object class from $C_1$, the inference task is to uncover all the corresponding instance(s) of the same class in a target image $I$. Notice that in this work we assume each feasible target image includes at least one object instance with respect to the class label of the one-shot query. We particularly focus on three key issues in training the underlying neural network and introduce new concepts to effectively perform one-shot object detection. We next describe the motivations, the reasoning, and the details of the proposed techniques.

**Non-local object proposals**   We denote the training dataset as $D$ with bounding box information over the class labels from $C_0$. In view that we adopt the Faster R-CNN architecture for object detection, the first issue arises essentially to examine whether the region proposals generated by RPN (Region Proposal Network) are suitable for one-shot object detection. Recall that the training of an RPN uses the information of the presence of bounding boxes over all object classes in each image. However, in our setting only those ground-truth boxes corresponding to the labels in $C_0$ can be accessed in learning the RPN. The constraint implies if a one-shot object class in $C_1$ is significantly *different* from any of those in $C_0$, the resulting RPN may not yield an expected set of proposals for detecting the corresponding object instances in a target image. To resolve this matter, we enrich the conv feature maps of interest with the non-local operation [25]. Again let $I$ be the target image and $p$ the query image patch. The conv feature maps used by the conventional RPN to generate the proposals are expressed by $\phi(I) \in \mathbb{R}^{N \times W_I \times H_I}$, while $\phi(p) \in \mathbb{R}^{N \times W_p \times H_p}$ represents the feature maps of patch $p$ from the same conv layer. Taking $\phi(p)$ as the input reference, the non-local operation is applied to $\phi(I)$ and results in a non-local block, $\psi(I; p) \in \mathbb{R}^{N \times W_I \times H_I}$. Analogously, we can derive the non-local block $\psi(p; I) \in \mathbb{R}^{N \times W_p \times H_p}$ using $\phi(I)$ as the input reference. The mutual non-local operations between $I$ and $p$ can indeed be thought of as performing co-attention. Finally,

we can represent the two extended conv feature maps by

$$F(I) = \phi(I) \oplus \psi(I; p) \in \mathbb{R}^{N \times W_I \times H_I} \quad \text{for target image } I, \tag{1}$$

$$F(p) = \phi(p) \oplus \psi(p; I) \in \mathbb{R}^{N \times W_p \times H_p} \quad \text{for image patch } p, \tag{2}$$

where $\oplus$ is the element-wise sum over the original features maps $\phi$ and the non-local block $\psi$. Since $F(I)$ comprises not only image features from the target image $I$ but also the weighted/attended features between $I$ and the query patch $p$, designing the RPN based on the extended features would learn to explore more information from the query patch $p$ and generate region proposals of better quality. In other words, the resulting non-local region proposals will be more appropriate for one-shot object detection.

**Squeeze and co-excitation** Besides linking the generation of region proposals with the given query patch, the co-attention mechanism realized by the non-local operation elegantly arranges the two sets of feature maps $F(I)$ and $F(p)$ for having the same number (*i.e.*, $N$) of channels. The relatedness between the two can be further explored by our proposed *squeeze-and-co-excitation* (SCE) technique such that the query $p$ can flexibly match a candidate proposal by adaptively re-weighting the importance distribution over the $N$ channels. Specifically, the squeeze step spatially summarizes each feature map with GAP (global average pooling), while the co-excitation functions as a bridge between $F(I)$ and $F(p)$ to simultaneously emphasize those feature channels that play crucial roles in evaluating the similarity measure. In between the squeeze layer and the co-excitation layer, we have two fc/MLP layers as in the design of an SE block [26]. We depict the SCE operation as follows.

$$\text{SCE}(F(p), F(I)) = \mathbf{w}, \quad \widetilde{F}(p) = \mathbf{w} \odot F(p), \quad \widetilde{F}(I) = \mathbf{w} \odot F(I), \tag{3}$$

where $\widetilde{F}(p)$ and $\widetilde{F}(I)$ are the re-weighted feature maps, $\mathbf{w} \in \mathbb{R}^N$ is the co-excitation vector, and $\odot$ denotes the element-wise product. With (3), the query patch $p$ can now be represented by

$$\mathbf{q} = \mathbf{w} \odot \text{GAP}(F(p)) = \text{GAP}(\widetilde{F}(p)) \in \mathbb{R}^N, \tag{4}$$

while the feature vector, say, $\mathbf{r}$ for a region proposal generated by RPN can be analogously computed, *i.e.*, performing spatially GAP over the corresponding cropped region of $\widetilde{F}(I)$.

**Proposal ranking** Assume that $K$ region proposals by RPN are chosen as possible candidates for object detection with respect to the query image patch $p$. ($K = 128$ in all experiments.) We design a two-layer MLP network $\mathcal{M}$, whose ending layer is a two-way softmax. In the training stage, we first annotate each of the $K$ proposals as foreground (label 1) or background (label 0) according to whether their IoU value with respect to the bounding-box ground truth is greater than $0.5$. We then consider a margin-based ranking loss to implicitly learn the desirable metric such that the most relevant proposals to the query $p$ would appear in the top portion of the ranking list. To this end, we concatenate for each proposal its feature vector $\mathbf{r}$ with the feature vector $\mathbf{q}$ from (4) to obtain a combined vector, denoted as $\mathbf{x} = [\mathbf{r}^\mathsf{T}; \mathbf{q}^\mathsf{T}]^\mathsf{T} \in \mathbb{R}^{2N}$, whose label $y$ is 1 if $\mathbf{r}$ corresponds to a foreground proposal, and 0, otherwise. We choose to construct $\mathcal{M}$ with layer dimensions distributed by $2N \rightarrow 8 \rightarrow 2$. Now let $s = \mathcal{M}(\mathbf{x})$ be the foreground probability predicted by $\mathcal{M}$ with respect to the query $\mathbf{q}$. We define the margin-based ranking loss by

$$\mathcal{L}_{\text{MR}}(\{\mathbf{x}_i\}) = \sum_{i=1}^{K} y_i \times \max\{m^+ - s_i, 0\} + (1 - y_i) \times \max\{s_i - m^-, 0\} + \Delta_i, \tag{5}$$

$$\Delta_i = \sum_{j=i+1}^{K} [y_i = y_j] \times \max\{|s_i - s_j| - m^-, 0\} + [y_i \neq y_j] \times \max\{m^+ - |s_i - s_j|, 0\}, \tag{6}$$

where $[\cdot]$ is the Iverson bracket, the margin $m^+$ is the expected probability lower bound for predicting a foreground proposal and $m^-$ is the expected upper bound for predicting a background proposal. In our implementation, we have set $m^+ = 0.7$ and $m^- = 0.3$ for all the experiments.

Finally, the total loss for learning the neural network architecture shown in Figure 1 to carry out one-shot object detection can be expressed by

$$\mathcal{L} = \mathcal{L}_{\text{CE}} + \mathcal{L}_{\text{Reg}} + \lambda \mathcal{L}_{\text{MR}}, \tag{7}$$

where the first two losses are respectively the cross entropy and regression losses of Faster R-CNN.

Table 1: Comparison of different few-shot detection methods on VOC in AP (%). 'Ours (725)' means our model is pre-trained on a reduced ImageNet dataset to prevent from foreseeing the unseen classes. Note that SiamFC, SiamRPN, and CompNet use all classes in their ImageNet pre-trained backbones.

| Method | Seen class | | | | | | | | | | | | | | | | | Unseen class | | | | |
|---|---|---|---|---|---|---|---|---|---|---|---|---|---|---|---|---|---|---|---|---|---|---|
| | plant | sofa | tv | car | bottle | boat | chair | person | bus | train | horse | bike | dog | bird | mbike | table | mAP | cow | sheep | cat | aero | mAP |
| SiamFC | 3.2 | 22.8 | 5.0 | 16.7 | 0.5 | 8.1 | 1.2 | 4.2 | 22.2 | 22.6 | 35.4 | 14.2 | 25.8 | 11.7 | 19.7 | 27.8 | 15.1 | 6.8 | 2.28 | 31.6 | 12.4 | 13.3 |
| SiamRPN | 1.9 | 15.7 | 4.5 | 12.8 | 1.0 | 1.1 | 6.1 | 8.7 | 7.9 | 6.9 | 17.4 | 17.8 | 20.5 | 7.2 | 18.5 | 5.1 | 9.6 | 15.9 | 15.7 | 21.7 | 3.5 | 14.2 |
| CompNet | 28.4 | 41.5 | **65.0** | 66.4 | 37.1 | 49.8 | **16.2** | 31.7 | 69.7 | 73.1 | 75.6 | 71.6 | 61.4 | 52.3 | 63.4 | 39.8 | 52.7 | 75.3 | 60.0 | 47.9 | 25.3 | 52.1 |
| Ours (725) | 24.9 | **50.1** | 58.8 | 64.3 | 32.9 | 48.9 | 14.2 | **53.2** | 71.5 | 74.7 | 74.0 | 66.3 | **75.7** | 61.5 | 68.5 | **42.7** | 55.1 | 78.0 | **61.9** | 72.0 | 43.5 | 63.8 |
| Ours (1k) | **30.0** | **54.9** | 64.1 | **66.7** | **40.1** | **54.1** | 14.7 | **60.9** | **77.5** | **78.3** | **77.9** | **73.2** | **80.5** | **70.8** | **72.4** | **46.2** | **60.1** | **83.9** | **67.1** | **75.6** | **46.2** | **68.2** |

## 4 Experiments

**Datasets and hyperparameters**    Following the previous work [22, 24], we train and evaluate our model on VOC and COCO benchmark datasets. For VOC, our model is trained on the union set of VOC 2007 train&val sets and VOC 2012 train&val sets, and is evaluated on VOC 2007 test set. For COCO, our model is trained on COCO 'train 2017' set and evaluated on COCO 'val 2017' set. Table 1 shows the splits of seen and unseen VOC classes, the same setting as [24]. For COCO, we use the same four splits over the 80 classes as [22], alternately taking three splits as seen classes and one split as unseen classes (see Figure 2). We train our models using SGD optimizer with momentum 0.9 for ten epochs, with batch size 128 on eight NVIDIA V100 GPUs in parallel. We use a learning rate starting with 0.01, and then decay it by a ratio 0.1 for every four epochs. We use $\lambda = 3$ in (7) for the margin-based ranking loss.

**Generating target and query pairs**    The target images are directly chosen from the datasets. To generate a query image for a target image, we adopt different generation procedures for different datasets. For VOC, we simply crop out the ground truth bounding boxes as the query image patches. For COCO, however, such a cropping procedure cannot simply be applied, since the cropped image patches might be either too small or too hard to identify even for humans. Therefore, we adopt a pre-trained Mask-RCNN [10] [1] to filter out the too small or too hard query image patches. Specifically, we only crop out the patches that are enclosed by the predicted bounding boxes of Mask R-CNN. During training, given a target image, we randomly sample a query image patch of a seen class that exists in the target image. During testing, to evaluate each class in a target image, we first randomly shuffle the query image patches of that class with a random seed of target image ID (the image ID is accessible in either VOC or COCO), then sample the first five query image patches, and finally average their AP scores. The shuffle procedure ensures that the sampled five query patches would be random and thus result in stable statistics for evaluation.

**ImageNet pre-training**    To ensure that our model does not 'foresee' the unseen classes, we pre-train our ResNet-50 backbone on a reduced training set of ImageNet from which we remove all COCO-related ImageNet classes by matching the WordNet synsets of ImageNet classes to COCO classes, resulting in 933, 052 images from the remaining 725 classes, while the original one contains 1, 284, 168 images of 1, 000 classes. Our pre-trained ResNet-50 achieves 75.8% (top-1) accuracy on the reduced ImageNet. Note that, by removing the COCO classes from ImageNet, we are also guaranteed to exclude the VOC classes from ImageNet.

**Baselines**    We choose the previous work that is closely related to our work as the baseline methods, each evaluated on different datasets: For VOC dataset, SiamFC [27], SiamRPN [28], and CompNet [24] are the baseline methods to be compared. CompNet builds on Faster R-CNN and replaces the conventional classifiers with the metric-based classifiers in both RPN and R-CNN, while SiamFC and SiamRPN (outperformed by CompNet) aim to solve the visual tracking problem instead of focusing on one-shot object detection. Note that SiamFC, SiamRPN, and CompNet do not remove unseen classes in their ImageNet pre-trained backbones, while ours removes both seen and unseen classes from the pre-trained backbone. For COCO, SiamMask [22] sets up a baseline performance on COCO dataset. SiamMask extends Mask R-CNN [10] with a feature pyramid network [29] and also replaces the conventional classifier with a metric-based classifier in R-CNN. We compare their model with ours on COCO dataset under the same setting.

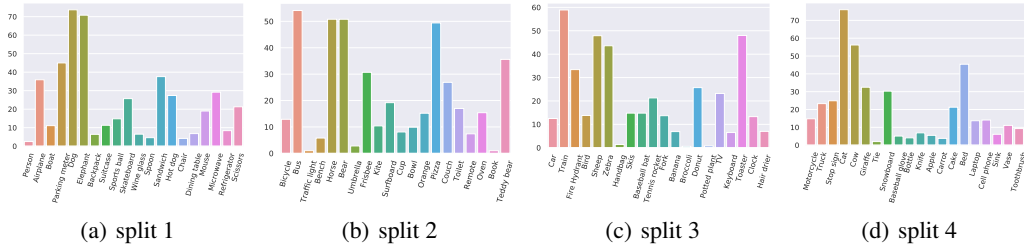

|  | (a) split 1 | (b) split 2 | (c) split 3 | (d) split 4 |

Figure 2: The AP50 (%) on different splits of COCO unseen classes. Each split is alternately used as unseen classes for evaluation, with the other three splits as seen classes for training.

Table 2: Evaluation on COCO val 2017 with respect to AP50 score (%).

| split | 1 | 2 | 3 | 4 | Average |
|---|---|---|---|---|---|
| SiamMask (seen) | 38.9 | 37.1 | 37.8 | 36.6 | 37.6 |
| Ours (seen) | 42.2 | 40.2 | 39.9 | 41.3 | 40.9 |
| SiamMask (unseen) | 15.3 | 17.6 | 17.4 | 17.0 | 16.8 |
| Ours (unseen) | 23.4 | 23.6 | 20.5 | 20.4 | 22.0 |

**Overall performance**  For VOC, Table 1 shows that our model using reduced ImageNet pre-trained backbone ('Ours (725)') still achieves better performance on both seen and unseen classes than the baseline methods. Furthermore, the performance significantly improves when we also adopt ImageNet pre-trained backbone with all 1000 classes ('Ours (1k)'). However, the unseen classes have better performance than seen classes due to the high variations in appearance of seen objects such as plant, bottle, and chair. For COCO, Table 2 also shows that our model achieves better performance than Siamese Mask-RCNN on both seen and unseen classes. Figure 2 further shows the fine-grained performance on each class; the artifact classes are the hardest to detect since they vary in textures and shapes, such as hand bag, book, and tie.

# 5   Ablation studies

**Co-attention, co-excitation, and margin-based ranking loss**  We investigate the contributions of different proposed modules and summarize the results in Table 3. First, the model without both co-attention (non-local RPN) and co-excitation (SCE) gets the worst performance. However, adding either non-local RPN or SCE significantly boosts the performance with an increase of $4.4/6.3$ mAP(%) and $8.2/9.8$ AP50(%) on VOC and COCO, respectively. Applying both modules further provides $1.8/0.9$ mAP(%) and $1.9/0.3$ AP50(%) performance gains. This implies that both co-attention and co-excitation are crucial to our method. The margin-based ranking loss also enhances the performance moderately, which means that margin-based ranking loss can still be helpful for learning the desirable metric.

**Visualizing the distribution of non-local object proposals**  To analyze the behavior of non-local object proposals, we visualize the distribution of proposals as a heatmap. Each pixel associates with a count that indicates how many region proposals cover that pixel. The final heatmap is then produced by normalizing the pixel count to a probability map. As shown in Figure 3, the mutual non-local

Table 3: Ablation study on co-attention (non-local object proposals), co-excitation (SCE), and margin-based ranking loss.

| Co-attention | Co-excitation | Margin loss | VOC mAP (%) | COCO AP50 (%) |
|---|---|---|---|---|
| ✓ | ✓ | ✓ | 57.0 | 23.6 |
|  | ✓ | ✓ | 54.2 | 21.7 |
| ✓ |  | ✓ | 56.1 | 23.3 |
| ✓ | ✓ |  | 51.6 | 22.4 |
|  |  | ✓ | 49.8 | 13.5 |

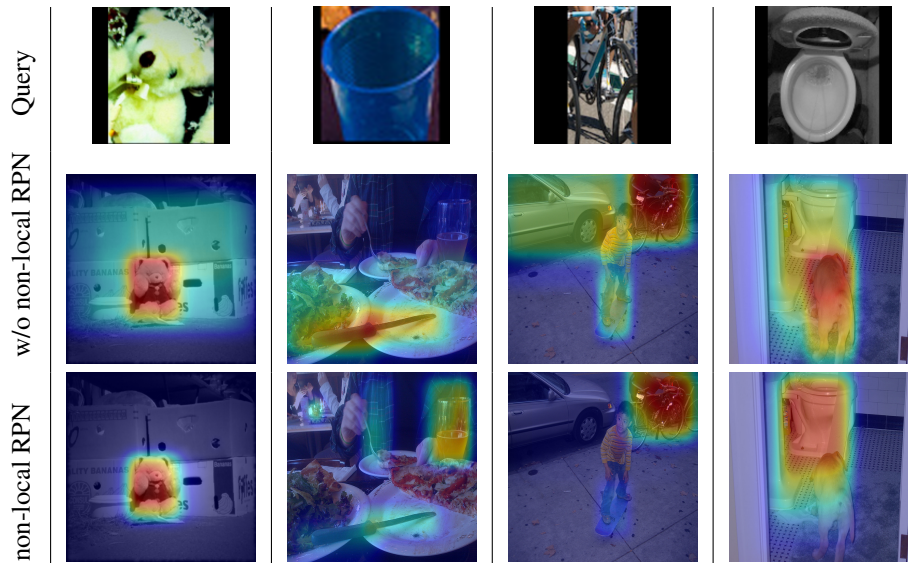

Figure 3: Non-local RPN is useful for attracting more proposals on the correct targets.

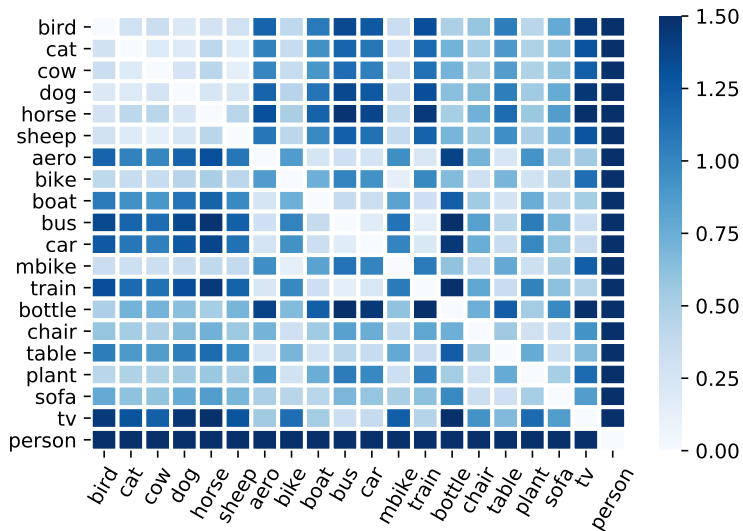

Figure 4: Visualization of class-to-class pairwise distances based on co-excitation weights.

features enable the RPN to generate proposals that focus more on the regions of both the target's and query's interest, and hence provide a co-attention effect.

**Visualizing the characteristics of co-excitation**   To analyze whether our proposed co-excitation mechanism learns a different weight distribution for each class, we collect all co-excitation weights for every query image during testing. Therefore, each class associates with a set of query images, and each of the query images associates a set of co-excitation weights. For each class, we average the co-excitation weights to a single vector. The visualization of class-to-class pairwise distances is then carried out by computing the pairwise Euclidean distance of the co-excitation weight vector of each class-to-class pair. Figure 4 clearly points out that our 'squeeze and co-excitation' module learns a meaningful weight distribution for each class. For example, the co-excitation weights of animal-related classes are closer to each other. A similar phenomenon can be observed for vehicle-related classes. On the other hand, the person class is far away from all the other classes, meaning that the person class is hard to share the common attributes either in texture or in shape with the other classes.

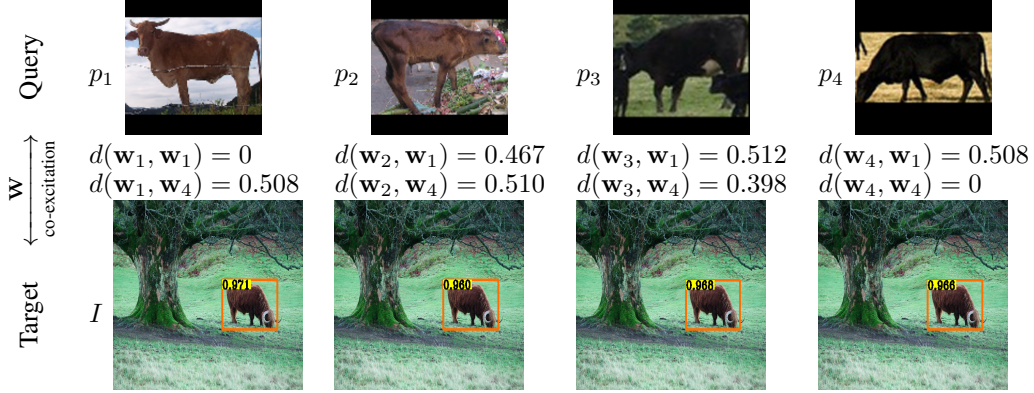

Figure 5: Query $p_i$ to the same $I$ results in the co-excitation $\mathbf{w}_i$. Both $\mathbf{w}_1$ and $\mathbf{w}_2$ should emphasize the color channels to detect the target instance in $I$, while $\mathbf{w}_3$ and $\mathbf{w}_4$ focus on those related to shape.

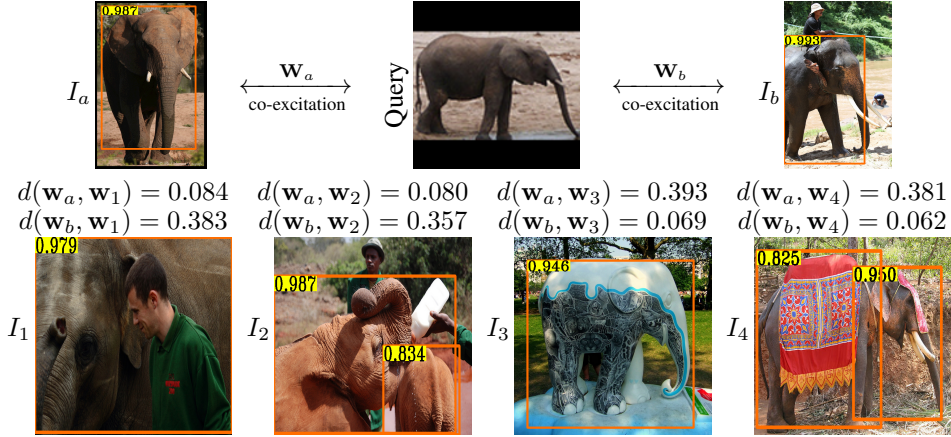

Figure 6: The same query $p$ to each different $I_x$ results in a co-excitation vector $\mathbf{w}_x$. Observe that $d(\mathbf{w}_a, \mathbf{w}_1), d(\mathbf{w}_a, \mathbf{w}_2) \ll d(\mathbf{w}_a, \mathbf{w}_3), d(\mathbf{w}_a, \mathbf{w}_4)$ (favoring texture features) and $d(\mathbf{w}_b, \mathbf{w}_1), d(\mathbf{w}_b, \mathbf{w}_2) \gg d(\mathbf{w}_b, \mathbf{w}_3), d(\mathbf{w}_b, \mathbf{w}_4)$ (favoring shape features).

**Analyzing the co-excitation mechanism** We consider two opposite cases. The first scenario is to use different image patches to query the same target image. Figure 5 shows that the cows in $p_1$ and $p_2$ share a similar color to the target instance in $I$, while the other two in $p_3$ and $p_4$ are of different colors to the target. A reasonable conclusion is that the former would emphasize the color features and the latter two the shape features so that each query can match to the target instance in $I$. The observation is supported by that $\mathbf{w}_2$ is closer to $\mathbf{w}_1$ than both $\mathbf{w}_3$ and $\mathbf{w}_4$. The second case is to use a same query $p$ for different target images. Analogously, in Figure 6, the distances between every two co-excitation vectors are insightful. In particular, the two sets of distance values suggest that the query $p$ to $I_1$ and $I_2$ would pay more attention to texture features, rather than the shape features as to $I_3$ and $I_4$.

## 6   Conclusion

In designing the proposed CoAE one-shot object detector, we have intentionally cast the learning formulation such that it does not solely rely on the label information of training data. Both the proposed co-attention and co-excitation techniques are to explore the correlated evidence revealed by the query-target pairs. Such information is generic and not heavily biased to the training data. As a result, the proposed method can yield non-local object proposals and uses the co-excitation operation to emphasize important features shared by both the query and the target images. The resulting one-shot object detector achieves state-of-the-art performances on two popular datasets. The future work will focus on generalizing our method for any $k$-shot ($k \geq 0$) object detection.

## Acknowledgement

This work was supported in part by the Ministry of Science and Technology (MOST), Taiwan under Grants 106-2221-E-007-080-MY3, 107-2218-E-007-047, 107-2634-F-001-002, and 108-2634-F-001-007. We are particularly grateful to the National Center for High-performance Computing (NCHC) for providing computational resources and facilities. The authors also like to thank Songhao Jia for insightful discussions on the implementation.

## Footnotes

[1]Mask R-CNN implementation: `https://github.com/matterport/Mask_RCNN`

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
