[Reviews · NeurIPS 2019]

Reviewer 1



The paper addresses a special case of few-shot object detection, subject to the following restrictions: 1. Only one unseen category is sought for in a target image at each time; 2. At least one object of the sought unseen category is assumed present in the target image; 3. Only one example of the unseen category can be used at each time. In my opinion, these conditions limit the facility of the stated task (and, therefore, the proposed method). There are (few) existing prior works addressing the more general problem of receiving any number of examples (down to one) of any number of unseen categories, and performing detection without assumption that the objects of specific categories are present in the target images. Examples of prior works include [20,21,23] (in the paper's bibliography) The description of the proposed model in the paper is not sufficiently clear. The diagram in Figure 1 contains only standard computation blocks (e.g. ResNet backbone, RPN, ROI align) and the data blobs, while the novel proposed computation layers are not drawn for some reason (except the elementwise addition and multiplication). Therefore, it is difficult to follow the description in the text, as the notation for these layers is missing from the diagram (e.g., the funcitons \psi, SCE). The second step - squeeze and co-exitation - based on the work [26]. Here its textual description contradicts the diagram: the vector of weights depends on both F(p) and F(I) in the text and only on F(p) in the diagram. Moreover, it is stated that the reweighed feature vector \tilde{F}(I) is the input to RPN, but in the diagram RPN receives the initial F(I) vector. I recommend the authors to synchronize their graphical description to the text and fill in the missing blocks and notation. The margin-based ranking loss is a standard one for binary classifier, and its presence makes sense. However, I am not sure its usage can be claimed as a significant contribution. The experimental section details on the (two existing) benchmarks on which the algorithm is tested. The reported performance is non-negligibly higher than the SOA, which points to the effectiveness of the proposed attention mechanism. The ablation study is satisfactory and exhibits the effectiveness of the proposed non-local RPN module. The literature review is accurate and is satisfactory. To summarize, I recon the proposed idea has a moderate importance for the template matching style algorithms in DNN era (e.g., conditional object detection).

Reviewer 2



POST-REBUTTAL COMMENTS Thanks for your response. Impact of conditional-RPN: Thanks for acknowledging that the impact of the conditional-RPN alone is not measured. Could you run an experiment using object-agnostic region-proposals for the final version? Form of SCE(): If I understand your response, then w = SCE(F(p), F(I)) does not use F(I) at all? If so, this notation is misleading, please fix it. Figure 1: Great, this will be more clear with an arrow coming from \tilde{F}(I). Hierarchy: You did not address my concerns about the unclear discussion of a "hierarchy". Ranking loss: You did not address my concerns about the combined ranking and classification losses. Your reason for using the same margin terms (to reduce the number of hyper-parameters) is quite weak. Nevertheless, I remain positive about the paper. The method seems highly effective compared to other methods that do not use fine-tuning. Please try to address my concerns for the final version. ORIGINAL REVIEW # Originality The idea is sufficiently novel. Related work is quite well covered. # Quality Overall, I found the experimental investigation to be of high quality. When training the RPN with co-attention, I imagine that only detections for the query class are used as positives. When training the RPN for the ablative experiment in which the co-attention module is disabled, are you sure to adopt all classes in C_0 as positive examples? Otherwise the training task may be too difficult, as the RPN is not aware of the query. One of the claims of the paper is that the RPN using co-attention is superior to a simple class-agnostic RPN that does not depend on the query image. It is confirmed qualitatively in Figure 3 that the proposed RPN focuses much more on the query object. However, this is rather anecdotal, and for an RPN the recall is usually more important than the precision. It would have been better if the authors compared to an RPN that did not depend on the query image p and was trained for all classes in C_0. They do show in ablative experiments that removing co-attention is worse, however this affects later stages as well as the RPN. To motivate the design of the architecture, it is hypothesized that allowing the network to depend on both the query patch p and the entire target image I allows it to infer the latent level of some class hierarchy, which disambiguates the one-shot task. This is an interesting way to define the task of one-shot detection. However, the hypothesis is not tested at all. To test this, there should be some experiments where e.g. the query image is a dog and the target image contains only a cat (latent class is mammal). The model is never trained to be aware of a class hierarchy. This makes the motivation rather weak. The ranking loss in equations 5 and 6 is mostly logical except: 1. The second term in equation 6 (for yi ≠ yj) could be improved. Using an absolute value here tries to ensure that si and sj are different when yi and yj are different. Instead, you should try to make the score of the positive example greater than the score of the negative example. This is more appropriate for a "ranking" loss. 2. Why use the same m+ and m– in equation 6 as in equation 5? These seem to be fundamentally different quantities. In equation 5, m+ and m– refer to the thresholds for positive and negative scores. In equation 6, m– means "maximum distance between scores with the same label" and m+ means "minimum distance between scores with different labels". 3. Are there existing works which use a similar loss function with a summation over all pairs? Could you add a citation? # Clarity The paper is mostly well written and pleasant to read. In the Faster-RCNN paper, they propose two strategies for training the RPN and the detection network with shared convolutional layers: alternating and joint. Which do you employ? The architecture of the output stage is not clear. 1. Do you output separate scores for the "proposal ranking" loss L_{ME} and classification loss L_{CE}, or are these two losses applied to the same scores? If you apply both losses to the same network output, then it seems redundant to combine a margin classification loss (first two terms of equation 5) and a cross-entropy classification loss? If you output two separate scores for ranking and classification, then why? It would be clearer if you added arguments to the loss functions in equation 7. 2. What architecture do you use for the bounding-box regression L_{reg}? The same architecture as the MLP for ranking? (but with 4 outputs instead of 2) The form of the squeeze function SCE() is not stated explicitly. It is simply stated that "the squeeze step spatially summarizes each feature map with GAP". While it is written that the channel-weight vector w is obtained from both F(p) and F(I) according to w = SCE(F(p), F(I)), Figure 1 seems to show that w does not depend on F(I). If this is the case, then "co-excitation" is very similar to regular squeeze-and-excitation. I do not understand the assumption that "the query object must not belong to any seen class at any level." Surely a class hierarchy would have a root node such as "object" or "thing". This class must be seen during training and any query object would belong to that class, contradicting the aforementioned statement. In general, lines 28-35 are unclear. It would be better if you introduce the idea of a class hierarchy before discussing it. The conclusion that "both co-attention and co-excitation are crucial" (line 226) is imprecise. It is true that each element, alone, was a crucial addition to the baseline. However, it seems that co-attention can achieve most of the performance alone, and adding co-excitation provides only a marginal improvement. There's no problem with this, it's interesting, but please be more precise. SiamFC and SiamRPN are designed to be scale-sensitive rather than scale-invariant. How did you apply these to global image search? Did you at least employ a (coarse) multi-scale search? # Significance The paper presents an effective and novel approach to the topical problem of one-shot detection. Two recent non-local methods are adapted for passing global information between branches. The paper is likely to be of wide interest. # Minor points Is it possible to visualize the spatial distribution of attention over the image patch p when obtaining F(p) from phi(p)? That is, how do different target images affect the representation of the query? The phrase "to use the non-local operation to explore the co-attention embodied in each query-target pair" in the abstract is not clear at all. The abstract should be easily understood. You should cite the journal version of [26]: https://ieeexplore.ieee.org/abstract/document/8701503 In Figure 1, tilde{F}(I) is not connected to anything? I believe it should be connected to the ROI-Align layer? (That is, the ROI-Align layer samples tilde{F}(I) using rectangles generated by the RPN?) The argument y_i is missing from the function L_{MR} in equation 5.

Reviewer 3



originality: as far as I know, the combinaison of Siamese network, nonlocal operation, co-excitation and margin-based loss in a 2 stage detector for 1-shot object detection is novel quality: the paper is technically sound. The experimental validation is is impressive (nice ablation study, strong results on 2 challenge,ging tasks). However, I have a major concern about the evaluation protocol: at test time the query image fed is from the classes present in target image, and performance is evaluated on that. This methodology uses information about test images, which is barely acceptable. It would have been better to see the results if the unseen classes don't even share the same meta-category as training classes. For eg: classes from vehicle metacategory can be kept for testing and animals metacategory used for training.  clarity: the paper reads very well. However, the introduction doesn't mention anything about margin based loss. Related works could talk about one shot detection instead of few shot detection and the intuition behind it. This section is too brief. Some citations are missing (eg: resnet paper) significance: the paper might have impact due to its originality and the quality of the experimental validation. post rebuttal remark: Regarding my comment on the evaluation protocol, I'm fine with the authors' response. After reading all the reviews and the rebuttal, I'm inclined to raise my rating to 7.

[Author Response · NeurIPS 2019]

** **Our method:** The proposed co-attention augments the feature maps (for RPN) of a target image $I$ with non-local
information w.r.t. the query patch $p$. The step leads to non-local object proposals. To determine the similarity between
the query patch and each proposal, we design a co-excitation mechanism to simultaneously re-weight and obtain the
feature maps $\tilde{F}(I)$ and $\tilde{F}(p)$. The co-excited feature representations are coupled with a margin-based ranking loss to
uncover instances similar to the query, no matter its category is seen or unseen. We emphasize that our method does not
require any model fine-tuning for carrying out the task of one-shot object detection over the unseen object categories.

** **Reviewer #1**

**1.1** *"The paper extends the previous work [24]... Other components of the detector are standard. I cannot name the..."*
While the design to generate non-local object proposals is new and its usefulness is illustrated in Figure 3, the other
main idea of our approach is to link the optimization of a margin-based ranking loss with co-excited features. As
demonstrated in the experiments (e.g., Figures 5 & 6), the formulation enables our detection method to uncover instances
of either a seen or an unseen class that resemble the query patch in different feature aspects such as shape and texture.

**1.2** *"The paper addresses a special case of few-shot object detection, subject to the following restrictions..."*
One-shot object detection is the most challenging one among all $k$-shot ($k \geq 1$) detection scenarios. The assumption
that the target image includes at least one object instance w.r.t. the class label of the query is assumed mainly for
training. In testing, although we have adopted the same tactic but it is for the sake of comparison with other techniques.
After all, in inference the detection results are subject to a unified score threshold (which is $0.5$ in our implementation).
The one-shot query in training is indeed to mimic the situation of one-shot (unseen class) object detection in inference.
Notice that in our formulation those single examples from unseen classes are provided only in inference and unlike [20,
21, 23], our method does not require any model fine-tuning for carrying out the one-shot (unseen) object detection.

**1.3** *"Here its textual description contradicts the diagram... reweighted feature vector $\widetilde{F}(I)$ is the input to RPN, but..."*
The reviewer may have missed the up and down arrows from the Channel Attention box in Figure 1 which conform
to $\text{SCE}(F(p), F(I)) = \mathbf{w}$. On lines 130–134, we explain that $F(I)$ is the input to RPN, as in Figure 1. The SCE
reweighted features $\tilde{F}(I)$ are instead coupled with the margin-based ranking loss. Also please see our method above.

**1.4** *"I believe [26] deserves more credit for the squeeze-and-excitation idea..."*
The SE block [26] is now a popular technique to re-weight the feature maps. In our writing we have explicitly stated
that our implementation of SCE follows [26]. However, unlike the SE block that works on feature maps from a single
source, the proposed co-excitation of SCE involves two streams of feature maps from two different input sources.

**1.5** We thank the reviewer for pointing out the mistake in misplacing "bus" and "person" in Table 1.

** **Reviewer #4** (Due to the limited space allowed, we have chosen to response to those most critical questions or concerns.)

**4.1** *"RPN using co-attention is superior to a simple class-agnostic RPN that does not depend on the query image..."*
We have provided qualitative results in Figure 3 and ablation evaluation in Table 3. As noticed by the reviewer, it is
difficult to quantitatively analyze the effect of RPN with or without performing the non-local operation w.r.t the query
patch in that the co-attention feature maps are used not only in RPN but also in the later stages of the network.

**4.2** *"The ranking loss in equations 5 and 6 is mostly logical except..."*
Since the first two terms in the RHS of (5) already enforce that foreground proposals would have larger scores than those
of the background ones, we emphasize only the score difference between each proposal pair depending on whether they
have the same class label. In (5) and (6), we have used the same $m^+$ and $m^-$ simply for reducing the number of margin
parameters from 4 to 2. We suspect that the all-pair ranking loss is not new but need to strive to find a relevant citation.

**4.3** *"The form of the squeeze function SCE() is not stated explicitly..."*
The squeeze step of SCE is performed only with $F(p)$. The co-excitation $\mathbf{w}$ involves two streams of feature maps (see
Figure 1) and re-weights $F(I)$ and $F(p)$ into $\tilde{F}(I)$ and $\tilde{F}(p)$. Our method learns to find an appropriate co-excitation
such that the margin-based ranking loss could prefer foreground proposals. Also see our response in 1.4.

**4.4** *"You should cite the journal version of [26]... In Figure 1, tilde{F}(I) is not connected to anything?..."*
Thank you and we will cite the journal version of [26]. We are sorry about the confusion in Figure 1. We have duplicated
$\tilde{F}(I)$ and positioned it after the RPN to indicate that the non-local proposals are represented with features from $\tilde{F}(I)$.

** **Reviewer #5**

**5.1** *"concern about the evaluation protocol: at test time the query image fed is from the classes present in target image"*
We agree with the reviewer that the current evaluation protocol can be improved. Still, it is adopted by most of the
related work about one-shot object detection, e.g., [22, 24]. As suggested by the reviewer, a more insightful evaluation
protocol, e.g., training from animal meta-categories and one-shot testing on vehicle meta-categories seems to pose a
very challenging problem of one-shot object detection, but we would attempt to carry out the suggested explorations
and include such experimental results for thoroughly testing the effectiveness of our proposed method.

[Meta-Review · NeurIPS 2019]

The reviewers questioned several components of these work and the authors' response resolved most of the issues. However, some of the raised questions were not addressed in the response (probably due to the lack of time and space). I would like to encourage the authors to carefully address those for the next revision.